# Initial Psychological Reactions to COVID-19 of Middle Adolescents in Portugal

**DOI:** 10.3390/ijerph20095705

**Published:** 2023-05-02

**Authors:** Rute Brites, Mauro Paulino, Sofia Brissos, Sofia Gabriel, Laura Alho, Mário R. Simões, Carlos F. Silva

**Affiliations:** 1Centro de Investigação em Psicologia (CIP), da Universidade Autónoma de Lisboa, 1169 Lisbon, Portugal; 2Mind Institute of Clinical and Forensic Psychology, 1990 Lisbon, Portugal; 3Center for Research in Neuropsychology and Cognitive and Behavioral Intervention (CINEICC), 3000 Coimbra, Portugal; 4Centro Hospitalar Psiquiátrico de Lisboa, 1749 Lisbon, Portugal; 5Faculdade de Psicologia e de Ciências da Educação, Universidade de Coimbra, 3000 Coimbra, Portugal; 6Departamento de Educação e Psicologia, Universidade de Aveiro, 3810 Aveiro, Portugal; 7William James Center for Research, ISPA & University of Aveiro, 3810 Aveiro, Portugal

**Keywords:** adolescents, mental health, COVID-19, pandemic

## Abstract

In its early stage, the COVID-19 pandemic and the subsequent public health measures brought several challenges to people in general, with adolescents being one of the most affected groups. To assess the psychological reactions of Portuguese adolescents in that early phase of the COVID-19 pandemic, we conducted an online survey that was filled by a sample of 340 (67.6% female and 32.4% male) middle adolescents (aged 16 and 17 years). Using the Impact Event Scale-Revised and the Depression, Anxiety, and Stress Scale, we found that most participants reported a normal score for depression, anxiety, and stress. However, 47.1% reported some level of pandemic-related traumatic distress, and 25.6% reported high severity values. The girls’ levels of depression, anxiety, stress, and traumatic distress were significantly higher than those of the boys. Regression models showed that gender, number of symptoms experienced in the past days, self-reported health status, and concern with family were significant predictors of these mental health indicators. Our findings underscore the need for future research on the long-term traumatic psychological impact of the COVID-19 pandemic in adolescents, and on the gender differences in this group. This will allow the development of strategies to identify and address at-risk adolescents, since the promotion of mental health and the prevention of pathology are imperative for the health of current and future generations.

## 1. Introduction

The coronavirus disease 2019 (COVID-19) pandemic, as a large-scale and uncontrollable event, had a major effect on the overall mental health of the population [1,2], especially before the vaccination process began.

Numerous studies report a consistent impact on individual mental health, with increases in people’s levels of stress, depression, anxiety [1,3,4,5,6,7], and post-traumatic stress disorder (PTSD) symptoms [1,8,9]. Evidence regarding the psychological impact of previous pandemics such as SARS or MERS highlight the traumatic potential of these events [10,11,12], with studies reporting increased values of PTSD symptoms in the survivors. With respect to the COVID-19 pandemic, studies show that the presence of uncertainty and anxiety due to the fear of contracting a highly infectious, potentially dangerous unknown disease allows us to consider the COVID-19 pandemic a traumatic event [13,14,15].

This mental health impact may have been particularly strong for adolescents [16,17]. Decreased face-to-face interactions, limitations on going out of the house, school difficulties, substantial changes in daily routine [18,19], as well as a perceived lack of safety, threat, and risk of contagion, disruption of continued learning, or parental quarantine due to infection [20] represent potentially impactful factors for adolescents.

Adolescence is a critical stage of personal, mental, and social development and can alter one’s life trajectory [21]. De France et al. [22] underline the importance of assessing this age group because they have been forced into an experience of social isolation at a stage when the salience of peer bonding is unparalleled, when there tend to be many conflicts with parents, and when there is a high risk that psychological symptomatology may generate individual vulnerability for life.

Adolescents are particularly vulnerable to stressful events as they have less experience and knowledge about how to cope with these occurrences [23]. This may increase the traumatic potential of the COVID-19 pandemic [24,25]. 

Some COVID-19-related studies have already shown this impact on adolescents. Results from a study by De France et al. [22] showed that the mental health of adolescents was affected due to the intensity of the global spread and the impact of the pandemic, as well as the societal disruptions caused by the pandemic prevention efforts, even when direct exposure to the virus itself was limited. According to Loades et al. [26], this group is more vulnerable to high rates of depression and anxiety during and after the end of imposed isolation, a finding confirmed by a systematic review by Jones et al. [27], who concluded that, overall, adolescents experienced higher rates of anxiety, depression, and stress. Temple et al. [28] also found that loneliness, stress, and economic challenges were linked to poor mental health and substance misuse. Magson et al. [29] demonstrated a significant increase in depressive and anxiety symptoms and a significant decrease in life satisfaction during the first COVID-19 wave, which was particularly pronounced among girls.

In fact, gender is a key factor in the overall state of youth mental health, both before and during the pandemic [30,31], with girls being more likely to experience anxiety and depressive symptoms [32,33]. In the same vein, Hu and Qian [34] reported a lower increase in emotional problems and a greater decrease in prosocial tendencies in boys compared with girls. However, some studies showed surprising results, describing an increase in depression symptoms only in boys, and an increase in anxiety symptoms only in girls [22].

According to the World Health Organization (WHO) [35], globally (and without considering the effects of the pandemic), an estimated 4.6% of adolescents between the ages of 15 and 19 suffer from an anxiety disorder, while depression is estimated to occur in 2.8% of adolescents in this age group. Thus, the increased psychological vulnerability in adolescents to the impact of a stressful event such as a pandemic, as well as the absence of consensual data on gender differences, leads us to the aim of this study: to assess the psychological reactions to the first wave of the COVID-19 outbreak in Portuguese middle adolescents. Thus, from the literature review and the purpose of this study, we formulated the following hypotheses:

**H1:** 
*Middle adolescents will show high levels of anxiety, depression, stress, and traumatic impact symptoms.*


**H2:** 
*Girls will present higher values of anxiety, depression, stress, and traumatic impact than boys.*


**H3:** 
*COVID-19-related variables (such as self-reported health status, number of symptoms, and concerns about self-infection and family infection) will be predictors of levels of anxiety, depression, stress, and traumatic impact.*


## 2. Materials and Methods

### 2.1. Participants

This study is a sub-analysis of a total of 11,014 participants who answered an online survey [1], of whom 405 were aged under 18. We excluded 65 due to their young age (according to Portuguese law, parental consent is needed for participants < 16 years old) and included reports from 340 participants aged 16 and 17 in the present analysis. Most participants were female (n = 230, 67.6%) and aged 17 years (n = 110, 56.5%), whereas 55% (n = 55) of the male adolescents were 16 years old. Almost the entire sample (n = 337, 99.1%) consisted of students, and 82% of the participants lived in households of 3 to 5 persons (n = 277). About 2/3 of the participants lived in an urban area (n = 216, 63.5%). In the period of data collection, all participants were in lockdown, attending classes in a distance learning system.

### 2.2. Measures

Besides psychological measures, the survey included socio-demographic questions (age, gender, education, household, and residence zone), a self-reported health status (ranging from 1, “poor health status”, to 5, “excellent health status”), and a health questionnaire related to COVID-19 [1], which included questions regarding belonging to a high-risk group (i.e., suffering from a previous chronic illness), the presence of COVID-19 symptoms (and how many), hospitalization due to COVID-19, concerns about infection and family infections (ranging from 1, “very worried”, to 4, “not worried at all”), and potential risk-contacts (yes or no).

Psychological measures included a scale evaluating the traumatic impact of a specific life event (the pandemic), the Impact Event Scale-Revised (IES-R) [36,37]. The scale has 22 items, answered through a 5-points Likert scale, ranging from 0 (“not at all”) to 4 (“extremely”). The total IES-R score was divided into minimal (scores 0–23), mild (24–32), moderate (33–36), and severe traumatic impact (≥37). This scale presents good internal consistency (original version α = 0.84 to 0.90; Portuguese version α = 0.96). In our study, the IES-R presented an excellent internal consistency (α = 0.93).

We also applied the Depression, Anxiety, and Stress Scale (DASS-21) [38,39]. This scale, with 21 items, assesses the presence of symptoms of depression, anxiety, and stress on a 4-point scale (ranging from 0, “did not apply to me at all”, to 3, “applied to me very much, or most of the time”). Each dimension is scored through 7 items. Total scores were multiplied by 2 to obtain DASS-42 scores. Depression scores were categorized as normal (0–9), mild (10–12), moderate (13–20), severe (21–27), or extremely severe (28–42). Anxiety scores were categorized as normal (0–6), mild (7–9), moderate (10–14), severe (15–19), or extremely severe (20–42). Stress scores were categorized as normal (0–10), mild (11–18), moderate (19–26), severe (27–34), or extremely severe (35–42). The DASS-21 has a well-established reliability, with Cronbach alpha values ranging from 0.83 to 0.94 for depression, 0.66 to 0.87 for anxiety, and 0.79 to 0.91 for stress [40]. In our study, the DASS-21 presented good reliability (α = 0.87 for depression; 0.84 for anxiety; 0.91 for stress).

### 2.3. Procedures

During the first wave of the COVID-19 outbreak (March 2020), the anonymous survey was launched online to respect the containment measures of lockdown and social isolation, and to reach a larger number of middle adolescents. The survey was launched on Facebook and Instagram and advertised in the media, being open to participation for a period of 72 h (24 March–27 March), one week after the first state of emergency declared by the Portuguese State and during the first global lockdown. The questionnaire briefing indicated that answers should be given considering the pandemic outbreak that was occurring in Portugal.

All ethical procedures regarding research with human subjects have been followed in accordance with the Declaration of Helsinki and the specific principle about research from the Code of Ethics of the Portuguese Psychological Association. All participants provided informed consent.

### 2.4. Data Analysis

Data analysis was performed using SPSS (version 24; IBM, SPSS Inc., Chicago, IL, USA). Descriptive statistics were calculated for all non-metric variables. Means (M) and standard deviations (SD) were calculated for metric variables (age, number of symptoms, stress, anxiety, depression, traumatic impact of COVID-19). Independent *t*-tests were conducted to examine the differences between groups. Hierarchical multiple regression was used to test the associations between the independent variables and the dependent variables (one at a time). The significance level for rejecting the null hypothesis was set at *p* < 0.05.

## 3. Results

### 3.1. COVID-19 and Health Measures

Concerning self-reported COVID-19 symptoms, most participants reported feeling well. The number of symptoms varied from 0 to 9, the average being less than one symptom (M = 0.98, SD = 1.32). This result was confirmed by the participants’ self-assessment of their current health status; on average, a value of 4 was reported (SD = 0.73, range 1–5). There were no participants belonging to a risk group (i.e., suffering from a chronic illness).

Considering the risk of infection, almost no one had undergone the screening test for COVID-19 (n = 4, 1.2%). Only four participants (1.2%) reported having been in direct contact with someone infected with SARS-CoV-2. Twenty-one participants (6.2%) reported having been in indirect contact, and 15.3% (n = 52) reported having been in contact with someone suspected to have COVID-19 or with infected materials. Most of the participants (n = 231, 67.9%) had been quarantined for the previous 14 days.

Most of these adolescents (n = 203, 59.7%) considered it unlikely that they would become infected with SARS-CoV-2, and less than 1% thought that it was possible to die from it. In general, they were satisfied with the publicly available information about the disease (n = 303, 89.1%) and believed they knew the virus’ routes of transmission (n = 331, 97.4%). Television and social networks were the main sources chosen for information. Adolescents were more concerned with the possibility of a relative becoming infected (n = 226, 66.5%) than with themselves.

### 3.2. Psychological Measures

The mean depression score during the initial phase of the first COVID-19 outbreak was 10.56 (SD = 10.57, range 0–42). Of all participants, 180 (52.9%) reported a normal score, 52 (15.3%) reported mild depression, 54 (15.9%) reported moderate depression, 21 (6.2%) reported severe depression, and 9.7% (n = 33) of the middle adolescents reported suffering from extremely severe depression.

For anxiety, the mean value was 7.21 (SD = 8.80, range 0–42). Most of the participants (n = 216, 63.5%) reported normal scores, 14 (4.1%) reported mild anxiety, 47 (13.8%) reported moderate anxiety, 29 (8.5%) reported severe anxiety, and 34 (10%) reported extremely severe anxiety.

The mean score for stress was 11.29 (SD = 11.63, range 0–42). Most of the adolescents had normal scores (n = 197, 57.9%), 18.2% (n = 62) reported mild stress, 10% (n = 34) reported moderated stress, and 9.7% (n = 33) reported severe stress. Only 4.1% (n = 14) reported extremely severe stress.

Regarding the traumatic impact of COVID-19, the mean score was 25.27 (SD = 18.06, range 0–88), compatible with mild symptomatology. In fact, about half of the participants (n = 180, 52.9%) reported minimal symptoms, and 21.4% (n = 73) reported mild (n = 60) or moderate (n = 13) symptoms; nevertheless, 25.6% (n = 87) reported severe symptoms, as is shown in Table 1.

### 3.3. Sociodemographic and Psychological Variables

Comparing female and male middle adolescents, the former reported significantly higher mean levels of depression, anxiety, and stress, and a significantly higher traumatic impact of COVID-19 (Table 2).

Considering the residence area (urban or rural), no differences were found regarding COVID-19 traumatic impact (*t* (333) = 1.73, *p* = 0.08), depression (*t* (333) = 1.14, *p* = 0.25), anxiety (*t* (333) = 0.65, *p* = 0.52), or stress (*t* (333) = 0.40, *p* = 0.69). There was, however, a tendency for adolescents living in rural areas (M = 22.91, SD = 17.56) to feel less of a traumatic impact than those living in urban areas (M = 26.48, SD = 18.34).

As mentioned, 25.6% of the adolescents reported a severe traumatic impact. Of these, 81.6% were girls, the majority (85%) were in high school, and they lived mostly in urban settings (68.6%). This number corresponds to the adolescents who were most concerned about family members contracting COVID-19, and the ones who reported the highest number of symptoms in the past 14 days and who rated their current health status the lowest. They were more likely to have been in indirect contact with confirmed or suspected cases of infection.

### 3.4. Health Status and Psychological Variables

The results for the associations between the psychological variables, the number of symptoms in the past 14 days, concerns about family, and self-reported health status are presented in Table 3.

Pearson correlations showed that the number of symptoms experienced in the previous 14 days was significantly associated with higher traumatic impact, depression, anxiety, and stress. A greater concern for family members was significantly associated with higher levels of trauma symptoms, but not with depression, anxiety, or stress.

Concern for family members was also negatively associated with the number of symptoms experienced in the past 14 days. Self-reported health status was negatively associated with traumatic impact, depression, anxiety, and stress. There was also a significant positive association between depression, anxiety, stress, and the traumatic impact of COVID-19.

### 3.5. Sociodemographic Variables, Health Status, and Psychological Variables

First, a two-stage hierarchical multiple regression was conducted with stress as a dependent variable. Gender was entered at stage one of the regression model to control for gender differences. The COVID-19 variables (number of symptoms in the past 14 days, self-reported health status, and concern for family members) were entered at stage two. The regression statistics are presented in Table 4.

The hierarchical multiple regression showed that, at stage one, gender contributed significantly to the regression model (*F* (1, 338) = 26.24, *p* < 0.001) and accounted for 7% of the variation in stress. Introducing the COVID-19 variables explained an additional 15% variation in stress, and this change in R^2^ was significant (*F* (1, 335) = 20.63, *p* < 0.001). Concern for family members was the only variable that was not a significant predictor of stress. Together, the four independent variables accounted for 22% of the variance in stress.

Next, a two-stage hierarchical multiple regression was conducted with anxiety as the dependent variable. Again, gender was entered at stage one of the regression model. The COVID-19 variables (number of symptoms in the past 14 days, self-reported health status, and concern for family members) were entered at the second stage. The statistics are presented in Table 5.

The hierarchical multiple regression showed that at stage one, gender contributed significantly to the regression model (*F* (1, 338) = 8.17, *p* < 0.01) and accounted for 2% of the variation in anxiety. The introduction of the COVID-19 variables explained an additional 18% of the variation in anxiety. This change in R^2^ was significant (*F* (1, 335) = 25.89, *p* < 0.001). When the four independent variables were included in stage two, neither gender nor concern for family members were significant predictors of anxiety. Together, the four independent variables accounted for 21% of the variance in anxiety.

A similar hierarchical multiple regression was then conducted, introducing depression as the dependent variable. Gender was entered at the first stage of the regression model, and at the second stage the same COVID-19 variables were entered. The regression results are presented in Table 6.

At the first stage of the regression model, the hierarchical multiple regression showed that gender contributed significantly to the regression model (*F* (1, 338) = 13.02, *p* < 0.001) and accounted for 4% of the variation in depression. The introduction of the COVID-19 variables explained a further 16% of the variation in depression, with a significant change in R^2^ (*F* (1, 335) = 22.61, *p* < 0.001). When the four independent variables were included in the model, concern for family members was the only variable that did not predict depression. The four independent variables, taken together, accounted for 20% of the variance in depression.

Finally, a last hierarchical multiple regression was performed, using traumatic impact as the dependent variable. In the first stage, gender was introduced as the predictor, and this was followed by the introduction of the four COVID-19 variables in the second stage. Table 7 presents the regression statistics.

The hierarchical multiple regression results showed that at stage one, gender contributed significantly to the regression model (*F* (1, 338) = 14.19, *p* < 0.01) and accounted for 4% of the variation in traumatic impact. The introduction of the COVID-19 variables explained an additional 7% of the variation in traumatic impact, with a significant R^2^ change (*F* (1, 335) = 8.63, *p* < 0.001). When the four independent variables were included in stage two, number of symptoms did not predict traumatic impact. Together, the independent variables accounted for 11% of the variance in traumatic impact.

## 4. Discussion

The present study aimed to examine the early psychological reactions, during the first wave of the COVID-19 outbreak, of Portuguese middle adolescents. Our findings show that, at this stage, the adolescents assessed themselves as healthy and were not particularly worried about being infected with SARS-CoV-2. They were more concerned about the possibility of their relatives becoming infected.

Regarding our first hypothesis (H1), “middle adolescents will show high levels of anxiety, depression, stress, and traumatic impact symptoms”, the results showed that most of the adolescents presented normal values, but a relevant percentage of them reported moderate to extremely severe values of depression, and around a quarter had stress values ranging from moderate to very high. In terms of anxiety, almost one fifth of the participants reported severe or extremely severe levels of anxiety. Finally, regarding traumatic impact, a quarter of the adolescents reported severe traumatic symptoms.

In general, these values are higher than those reported by the WHO [35] (2.8% for depression and 4.6% for anxiety in this age group) or by Correia-Santos et al. [41], who reported a 17.4% rate of PTSD symptoms in Portuguese adolescents. This enables us to confirm H1. Our results are in line with previous studies on the early psychological impact of the pandemic, which have reported decreased mental health [1,3,4,6,7,8,9], even in adolescents [9,17,33,42].

Gu et al. [43] have shown that exposure to social media leads to fear, anxiety, and depression. That exposure is markedly high in adolescents, which may help explain the results we obtained. On the other hand, it is possible that the adolescents’ concerns revolved around the economic situation [44,45], and especially related to school closures [46,47]. Frustration is another known consequence of confinement, having been shown to be elevated in adolescents during the COVID-19 pandemic [48] and to be related to higher levels of stress, depression, or anxiety [49].

Research has also highlighted the consequences of the societally forced disruptions caused by lockdown and isolation measures [22,26], including its impact on adolescents. Intimacy and closeness decrease when communication is only online, and although the online context is useful for socialization, the internet alone does not compensate for or replace person-to-person interaction [48]. The repercussions of social isolation could already have manifested at the very beginning of the lockdown, explaining the traumatic impact and symptoms of anxiety, depression, and stress.

Regarding the second hypothesis (H2), “girls will present higher values of anxiety, depression, stress and traumatic impact than boys”, the data allow its confirmation, since girls did indeed report higher scores than boys. This pattern of results is consistent with those previously reported in the literature indicating more pronounced symptoms and a more severe psychological impact in females, both in the adolescent population [32,33] and in adults [1,9,50]. Several studies have described a greater vulnerability of females to mental health issues [29,51,52,53], findings that are supported by the present data.

Finally, our third hypothesis (H3) stated that “COVID-19-related variables (self-reported health status, number of symptoms, and concerns about self-infection and family infection) will be predictors of levels of anxiety, depression, stress, and traumatic impact”. The data allow for its partial confirmation, as the regression values showed that being female, having more COVID-19 symptoms in the past 14 days, and having a poor self-reported health status were associated with higher depression and stress symptoms. Gender, self-reported health status, and concern for family members significantly predicted traumatic symptoms. The number of symptoms in the past 14 days, as well as the self-reported health status, were significant predictors of anxiety, but gender was not.

Since adolescents have a propensity to regard themselves as invulnerable [48], we would not expect such an important impact right from the beginning of the pandemic, even if recent studies showed a significant impact on the mental health of adolescents [22,27]. Some early reports from the first wave of the outbreak in China showed that the youth were fearful of what was happening [54] and that fear of COVID-19 increased with age [55], findings that may help explain our results regarding the impact of COVID-19-related variables.

Additionally, most of these adolescents lived in urban areas (63.5%, n = 216), which had, at the beginning of the pandemic, much higher rates of infection than rural areas. Symptoms of depression and anxiety are usually higher in extremely infected areas [56,57], where the elevated fear of infection and the perception of life-threatening danger impacts significantly on mental health [58]. In the early phase of the pandemic, the rural environment presented a lower risk of contagion, given the ease of physical distancing and the lack of cases of infected individuals. In a study by Duan et al. [52], living in an urban area was shown to be a risk factor for the heightened anxiety experienced by teenagers.

On the other hand, these adolescents were preoccupied with the possibility of a relative becoming infected (66.5%, n = 226), and a greater preoccupation was significantly associated with higher levels of traumatic impact. The fear of infection, illness, and death of family members due to the pandemic has been documented in other investigations with adolescents [32,56].

In the specific case of anxiety, where gender was no longer a significant predictor, it is possible that the levels displayed by the adolescents were directly associated with pandemic COVID-19 (a COVID-19-related anxiety), which would explain why only the number of symptoms in the past few days and the self-reported health status would predict anxiety, and why gender would not. Despite this, some studies have reported higher levels of COVID-19-related anxiety in girls [59,60], and this merit further investigation in the future.

## 5. Conclusions

The present study contributes to a growing body of evidence suggesting that the COVID-19 pandemic had an enormous impact on the mental health of middle adolescents. Although the worst is over, the long-term consequences of the pandemic and the measures imposed are not yet fully known. However, we do know that an earlier age of onset of mental disorders is associated with an increased risk of developing comorbidities and the persistence of mental health disorders into midlife [61]. Mental health difficulties that develop during adolescence can create lifelong vulnerabilities to mental health problems [62], and this confirms the need for the close monitoring of adolescents whose psychological response to the pandemic appears to have been significant. Some post-pandemic studies already provide a glimpse of this impact, with high levels of stress predicting problematic smartphone use and Internet gaming disorders [63]. If adolescence was already a complex and challenging phase, it is possible that the pandemic has increased the vulnerability of these young people, and this will require close monitoring and an individualized response from health entities.

## 6. Limitations and Future Research

Despite the relevance of our main findings, our study has some limitations that must be addressed. We used self-reported measures, which are more prone to bias, and a cross-sectional study design, which prevents us from drawing any conclusions about causality between variables. In addition, it is not possible to fully attribute the reported symptoms to the COVID-19 pandemic, as there were no pre-pandemic prevalence values, and we do not know whether these adolescents had previous symptoms or diagnoses.

Research at an early pandemic stage is important [50,64], and early suffering cannot be devalued. However, this study was conducted during the first month of the pandemic, when nothing was known about the severity of the disease and no vaccine or treatment was available. These results were obtained during a phase when fear of the unknown and death reigned [65,66]. It is therefore also possible to hypothesize that the symptoms described here were natural initial reactions to an unknown event. This possibility underlines the importance of long-term studies that can determine the prevalence of symptoms caused by the impact of the pandemic after it has ended. The finding that COVID-19 seems to impact girls differently also demonstrates the need to further explore the mechanisms underlying the psychological reactions of adolescents to COVID-19 and to develop specific strategies according to different gender and age groups.

## Figures and Tables

**Table 1 ijerph-20-05705-t001:** Descriptive statistics for mental health outcomes (depression, anxiety, stress, and traumatic impact) (N = 340).

Scale	Mean (SD)	Severity				
		Normal	Mild	Moderate	Severe	Extremely Severe
Depression	10.56 (10.57)	52.9%	15.3%	15.9%	6.2%	9.7%
Anxiety	7.21 (8.80)	63.5%	4.1%	13.8%	8.5%	10%
Stress	11.29 (11.63)	57.9%	18.2%	10%	9.7%	4.1%
Traumatic impact of COVID-19	25.27 (18.06)	52.9%	17.6%	3.8%	25.6%	0%

**Table 2 ijerph-20-05705-t002:** Associations between mental health outcomes (depression, anxiety, stress, and traumatic impact), number of symptoms in the past 14 days, concern about family, and self-reported health status (N = 340).

	1	2	3	4	5	6	7
1. Depression	–	0.81 **	0.71 **	0.64 **	0.34 **	−0.06	−0.36 **
2. Stress		–	0.80 **	0.75 **	0.27 **	−0.10	−0.39 **
3. Anxiety			–	0.68 **	0.32 **	−0.10	−0.40 **
4. Traumatic impact				–	0.16 **	−0.16 **	−0.25 **
5. Number of symptoms (past 14 days)					–	−0.10	−0.33 **
6. Concern about family						–	0.08
7. Self-reported health status							–

** *p* < 0.01.

**Table 3 ijerph-20-05705-t003:** Differences in psychological variables between female and male participants using *t*-tests (N = 340).

Score	Group	M	SD	Difference (*t*-Test)
Depression	Boys	7.62	10.14	*t* (338) = −3.61 ***
	Girls	11.97	10.51	
Anxiety	Boys	5.26	7.84	*t* (245.89) = −3.01 **
	Girls	8.14	9.09	
Stress	Boys	6.78	9.52	*t* (263.68) = −5.54 ***
	Girls	13.44	11.94	
Traumatic impact of COVID-19	Boys	20.04	17.58	*t* (338) = −3.77 ***
	Girls	27.77	17.78	

M = mean; SD = standard deviation; ** *p* < 0.05, *** *p* < 0.001.

**Table 4 ijerph-20-05705-t004:** Summary of hierarchical regression analysis for variables predicting stress (N = 340).

Variable	*B*	*t*	*R*	*R^2^*	∆*R^2^*
Step 1			0.27	0.07	0.07
Gender	0.27	5.12 ***			
Step 2			0.47	0.22	0.15
Gender	0.20	4.12 ***			
Number of symptoms	0.14	2.79 **			
Self-reported health status	−0.31	−5.95 ***			
Concern for family members	−0.05	−0.93			

** *p* < 0.01, *** *p* < 0.001.

**Table 5 ijerph-20-05705-t005:** Summary of hierarchical regression analysis for variables predicting anxiety (N = 340).

Variable	*B*	*t*	*R*	*R^2^*	∆*R^2^*
Step 1			0.15	0.02	0.02
Gender	0.15	2.86 **			
Step 2			0.46	0.21	0.18
Gender	0.08	1.60			
Number of symptoms	0.20	3.94 ***			
Self-reported health status	−0.31	−6.05 ***			
Concern for family members	−0.05	−1.04			

** *p* < 0.01, *** *p* < 0.001.

**Table 6 ijerph-20-05705-t006:** Summary of hierarchical regression analysis for variables predicting depression (N = 340).

Variable	*B*	*t*	*R*	*R^2^*	∆*R^2^*
Step 1			0.19	0.04	0.04
Gender	0.19	3.61 ***			
Step 2			0.45	0.20	0.16
Gender	0.13	2.51 *			
Number of symptoms	0.24	4.53 ***			
Self-reported health status	−0.26	−5.05 ***			
Concern for family members	−0.01	−0.20			

* *p* < 0.05, *** *p* < 0.001.

**Table 7 ijerph-20-05705-t007:** Summary of hierarchical regression analysis for variables predicting traumatic impact (N = 340).

Variable	*B*	*t*	*R*	*R^2^*	∆*R^2^*
Step 1			0.20	0.04	0.04
Gender	0.20	3.77 ***			
Step 2			0.33	0.11	0.07
Gender	0.16	2.96 **			
Number of symptoms	0.06	1.11			
Self-reported health status	−0.20	−3.56 ***			
Concern for family members	−0.13	−2.41 *			

* *p* < 0.05, ** *p* < 0.01, *** *p* < 0.001.

## Data Availability

The data that support the findings of this study are available from the corresponding author, [RB], upon reasonable request.

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
