# Peer review of "Initial Psychological Reactions to COVID-19 of Middle Adolescents in Portugal"

_ijerph, 2023, doi:10.3390/ijerph20095705_

Round 1

Reviewer 1 Report

Initial Psychological Reactions to COVID-19 on Adolescents in Portugal

ijerph-2339287

I thank you for the opportunity to review your manuscript and congratulate you on the study you carried out.

The data collected was of great interest at a crucial time, but I understand that this information should have been shared earlier. The authors have not shown the relevance of generating this work now, when the pandemic is already under control and the work cannot offer data on the evolution of the study variables. 

On the other hand, as the authors acknowledge in the limitations, not being able to establish causal relationships and not having prior measurements compromises the value of the findings.

The authors can review if they have other variables of interest that could have been collected in this information gathering and that may have an impact today. If so, I invite you to make a new submission.

Author Response

We appreciate the reviewers’ contribution on our study, and we thank them for it.  We changed the manuscript following reviewers’ comments, which are answered below. Changes in the text were made using Track Changes option.

A: Thank you for your comment. In fact, we would have liked the data to have been available sooner, but constraints associated with the operation of the journals and beyond our control did not allow this. Still, we believe that data concerning an initial moment of reaction to a new, stressful, and unfamiliar situation will always be useful. Unpredictability is increasingly part of our daily lives, and it is necessary to have as much data as possible in identical or parallel situations in the future. In terms of public health, knowing which groups are more vulnerable or potentially at higher risk can be important, regarding prevention, but also in reacting to new stressful events.

Reviewer 2 Report

The article presents a study that aimed to investigate the impact of the COVID-19 pandemic on the mental health of adolescents. The study utilized a variety of measures, including psychological measures, socio-demographic questions, and a health questionnaire related to COVID-19, to assess the mental health status of the participants and their perception of the pandemic. The article needs improvement:

11. absence of figures or visual aids to present the data, research model figures?. enhance the article's readability and made it easier for the reader to understand the results. Additionally, the article could have used structural equation modeling (SEM), a more robust statistical technique than hierarchical multiple regression, to analyze the data. The article would have benefitted from providing some justification for the use of hierarchical multiple regression instead of SEM and explaining why it was sufficient for the study.

2.2. the article could have included relevant literature to provide more context and support the findings.

The impacts of academic stress on college students' problematic smartphone use and Internet gaming disorder under the background of neijuan: Hierarchical regressions with mediational analysis on escape and coping motives. Front Psychiatry.

Gu, X., Obrenovic, B., & Fu, W. (2023). Empirical Study on Social Media Exposure and Fear as Drivers of Anxiety and Depression during the COVID-19 Pandemic. Sustainability, 15(6), 5312.

Godinic, D., Obrenovic, B., & Khudaykulov, A. (2020). Effects of Economic Uncertainty on Mental Health in the COVID-19 Pandemic Context: Social Identity Disturbance, Job Uncertainty and Psychological Well-Being Model. International Journal of Innovation and Economic Development, 6(1), 61-74.

Gerbersgagen, M. & I, S.A.S. (2023). Managers’ Lived Experience with Technology in the Mortgage Industry, During the COVID-19 Pandemic. International Journal of Management Science and Business Administration, 9(3), 7-15.

33. discussion section of the article is quite lengthy and could benefit from restructuring. The discussion should be focused on interpreting the results and explaining their implications. The authors could have used subheadings or bullet points to organize the discussion and make it easier for the reader to follow. And seperate limitations and conclusion

44 . The article did not explicitly state any hypotheses or research questions that the study aimed to answer. The absence of hypotheses can make it difficult for the reader to understand the research goals and the significance of the findings. It is essential to have well-defined hypotheses or research questions to guide the study and to provide a clear focus for the research.

Please modify accoridngly and provide replies.

Author Response

We appreciate the reviewers’ contribution on our study, and we thank them for it.  We changed the manuscript following reviewers’ comments, which are answered below. Changes in the text were made using Track Changes option.

1.absence of figures or visual aids to present the data, research model figures?. enhance the article's readability and made it easier for the reader to understand the results.

A: thank you. we understand your point of view. however, we felt that since this is an uncomplicated study model, the tables are sufficient to present the results.

Additionally, the article could have used structural equation modeling (SEM), a more robust statistical technique than hierarchical multiple regression, to analyze the data. The article would have benefitted from providing some justification for the use of hierarchical multiple regression instead of SEM and explaining why it was sufficient for the study.

A: Thank you for your question. We chose regression analysis for several reasons: 1) we did not have a complex model to test; 2) our goal was to test the relationship between the independent variables and the dependent variables one at a time; 3) since we applied widely tested measures used with this population, we did not consider it necessary to do a confirmatory analysis of the instrument. We added a short explanatory sentence in the Data Analysis section.

2.2. Thank you for your help. We have referred to some of the articles indicated below.

33. We really appreciate the suggestion. We have rewritten the discussion to meet the reviewers' suggestion. We added the conclusion and separated the limitations and future research.

44. 

A: We thank you for the request. In addition to the study objective that was already described, we added three hypotheses:

H1: Middle adolescents will show high levels of anxiety, depression, stress, and traumatic impact symptoms.

H2: Girls will present higher values of anxiety, depression, stress and traumatic than boys.

H3: COVID-related variables (such as self-reported health status, number of symptoms, and concerns about self-infection and family infection) will be predictors of levels of anxiety, depression, stress, and traumatic impact.

Reviewer 3 Report

This manuscript is well written and in the scope of this journal however the authors need to work on these few suggestions. 

1. WHO describes adolescence at the age between 11-19, but the subjects in this study are aged 16,17 years, but the authors still use adolescence though the age is skewed. I suggest the authors find an appropriate classification for their sample. 

2. Line 104: Can authors clarify what they mean by ''high risk group''

3. Line 156: Authors state that almost no one had done the screening test but they state (n=336, 98.7%) which is nothing close to null. Can they work on that. 

4. Line 157: Can authors state percentages to the number given? 

6. Line 160: Can authors add percentages in the text. 

7. In table 3, the authors use M for mean. I suggest authors use a more standard way of stating the mean in tables, because M can also mean Median. 

8. In Line 273-291, the authors summarize the results. After that they seem to discuss the results without any direct link to the findings of this study. I suggest in paragraphs after 291, author discuss the findings of the study making direct reference to the study. This will make readers appreciate the information better. -Specifically the paragraphs with line 306 to 318. 

9. Line 320, authors are required to state values and/or percentages for adolescents living in urban areas etc. 

10. Line: Kindly state values/percentages with reference to your data. 

11. Line 336-338. Authors didnt state into which context this information fits with respect to their results. Kindly do so. 

12. Line 349-350: Check sentence 

13. Clear conclusion is needed for this work. 

Author Response

We appreciate the reviewers’ contribution on our study, and we thank them for it.  We changed the manuscript following reviewers’ comments, which are answered below. Changes in the text were made using Track Changes option.

1. Thank you for this remark. We classified this group as middle adolescents.

2. We appreciate the request. We clarified in the text that these were people with chronic diseases.

3. Thank you for the observation. We presented in the text the percentage of people who did not take the test, but we realize that it is misleading. So, we replaced that value with the percentage of people who took the test.

4. Thank you for the suggestion. We added the percentage (1.2%).

6. Thank you. We added the percentage (67.9%).

7. We followed APA style (7th edition), where mean is represented by M and median by Mdn. however, we have added the meaning of the symbols in the caption of the table. We also added the symbols in the Data Analysis Section.

8. We thank you for this suggestion. We have rewritten the discussion to meet the reviewers' suggestion.

9. Thank you for the indication. This value was shown in the Participants section, but we put it back next to the indicated sentence.

10. Thank you for the suggestion. This value was presented in the Participants section, but we added it next to the indicated sentence.

11. Thank you. In fact, we failed to adequately convey our idea that the fact that the 1st diagnostic criterion of PTSD (criterion A) might be present in this sample could be a future warning factor. However, since the overall conclusion points out the importance of monitoring, we have removed the sentence.

12. Thank you. We did it. We replaced the sentence for: “Taken together, our results demonstrate the obvious impact of the pandemic and its circumstances on adolescent mental health.”

13. We appreciate the remark. We added a Conclusion section.

Round 2

Reviewer 1 Report

I congratulate the authors for the arrangements carried out.

The new version of the manuscript should correct some typographical errors and place the conclusion before the limitations.

Author Response

We appreciate the reviewer's acknowledgement of the effort we have made in modifying the text, which has, in fact, increased its scientific quality. We have revised the entire text, made minor changes related to verb tenses and other typos, and as requested, placed the Conclusion section before the Limitations.